# MbICE1 Confers Drought and Cold Tolerance through Up-Regulating Antioxidant Capacity and Stress-Resistant Genes in *Arabidopsis thaliana*

**DOI:** 10.3390/ijms232416072

**Published:** 2022-12-16

**Authors:** Yadong Duan, Jiaxin Han, Baitao Guo, Wenbo Zhao, Shuang Zhou, Chunwei Zhou, Lei Zhang, Xingguo Li, Deguo Han

**Affiliations:** 1Institute of Rural Revitalization Science and Technology, Heilongjiang Academy of Agricultural Sciences, Harbin 150028, China; 2Northeast Institute of Geography and Agroecology, Chinese Academy of Sciences, Harbin 150081, China; 3Huma Cold Temperate Zone Experimental Station of Conservation and Utilization of Wild Plant Germplasm Resources, Daxing’anling 165000, China; 4Key Laboratory of Biology and Genetic Improvement of Horticultural Crops (Northeast Region), Ministry of Agriculture and Rural Affairs, National-Local Joint Engineering Research Center for Development and Utilization of Small Fruits in Cold Regions, College of Horticulture & Landscape Architecture, Northeast Agricultural University, Harbin 150030, China

**Keywords:** *Malus baccata* (L.) Borkh, *MbICE1*, drought stress, cold stress

## Abstract

*Malus baccata* (L.) Borkh is an apple rootstock with good drought and cold resistance. The ICE gene is a key factor in the molecular mechanisms of plant drought and cold resistance. In the present research, the function of drought- and cold-induced *MbICE1* of *Malus baccata* was investigated in *Arabidopsis*. According to GFP fluorescence images, MbICE1 was determined to be a nuclear protein. The *MbICE1* was transferred to *Arabidopsis*, showing enhanced tolerance to drought and cold stresses. Under drought and cold treatments, the transgenic *Arabidopsis* had higher chlorophyll content and free proline content than WT plants, but the Malondialdehyde (MDA) content and electrolyte leakage (EL) were lower than those of WT plants. In addition, drought and cold led to a large accumulation of ROS (H_2_O_2_ and O^2−^) content in *Arabidopsis*, while overexpression of *MbICE1* enhanced the antioxidant enzyme activity in *Arabidopsis* and improved the plant’s resistance to stresses. Moreover, the accumulation of *MbICE1* promoted the expression of *AtCBF1*, *AtCBF2*, *AtCBF3*, *AtCOR15a*, *AtCOR47* and *AtKIN1* genes in *Arabidopsis*. These data indicate that *MbICE1* is a key regulator of drought and cold and can be used as a backup gene for breeding *Malus* rootstocks.

## 1. Introduction

Abiotic stresses are the main factors affecting agricultural production and include salinization as well as temperature and water stress; these can destroy plant cell homeostasis, affect cell division, and inhibit plant growth [1,2]. Cold and drought are the main abiotic stress factors involved in horticultural plant yield reduction in temperate regions [3]. When plants encounter cold and drought stresses, their physiological and biochemical levels undergo a series of changes to maintain the survival of the plants [4,5]. Reactive oxygen species (H_2_O_2_ and O^2−^) are produced continuously during plant growth and development. Under normal circumstances, the plant antioxidant enzyme system (superoxide dismutase, peroxidase and catalase) can remove ROS in time to maintain a dynamic balance, so it will not affect the metabolism of the plant. However, drought and cold stresses can lead to the accumulation of ROS in plants, break the dynamic balance, and destroy cells due to lipid peroxidation of cell membranes. MDA, as a product of peroxidation, can reflect the degree of plant damage, as one of the indicators of plant tolerance to drought and cold stresses. When stress signals are sensed by plants, stress-related functional proteins and regulatory factors are activated, thus activating the molecular regulatory pathways of plant response to stress and improving plant adaptability to adverse environments [6]. Among many stress regulation pathways, the ICE-CBF-COR pathway is the focus of researchers. As the upstream gene of this pathway, ICE is crucial to the whole pathway.

ICE1 (Inducer of CBF expression 1) is a bHLH TF containing a bHLH domain, which is numbered bHLH116 [7]. After sensing the stress signal, ICE1 can directly combine MYC (CANNTG) elements in the CBF3/DREB1A promoter to induce and activate the expression of stress-related genes, thereby improving the adaptability of overexpression plants to stresses [8]. Post-transcriptional control, such as ubiquitination and phosphorylation, are important for the activity of ICE1.

Previous research has found that the stability of the ICE1 protein is regulated by a pair of genes, including high expression of osmotically responsive gene 1 (HOS1) and SIZ1 (SUMO E3 ligase), with opposite regulatory effects. Under cold stress, HOS1 is a negative regulator, which degrades ICE1 protein through ubiquitination [9]. SIZ1 improves the stability of ICE1, inhibits the degradation of ICE1 and positively regulates the activity of ICE1 [10]. In addition, the protein kinase Open Stomata 1 (OST1) in the ABA core signaling pathway positively regulates cold stress by phosphorylating the ser278 site of ICE1 to keep ICE1 protein in a stable state [11]. Additionally, ICE1 can inhibit the expression of the *AtGA3ox4* gene and regulate the synthesis of endogenous GA, thus affecting the flowering period, the fertility of pollen, and the development of embryos and seeds [12].

ICE transcription factors have been found to regulate the response of abiotic stresses in many plants, such as eggplant, *Zea mays*, rice, *Isatis tinctoria*, *Phalaenopsis aphrodite*, *Eucalyptus camaldulensis*, tea tree, *Vitis vinifera*, and banana [13,14,15,16,17,18,19,20,21]. *ZjICE1* positively regulates abiotic stress in *Zoysia japonica*. Overexpression of *ZjICE1* enhances the ability to scavenge ROS and effectively maintains the oxidation–antioxidation homeostasis, so that plants can grow better under cold, drought, and salt stresses [22]. Overexpression of *HbICE2* in *Arabidopsis* shows that jasmonic acid can reduce the inhibition of *HbJAZ1/12* on *HbICE2*, improve the binding of *HbICE2* and *HbCBF1*, and thus enhance the cold resistance of *Hevea brasiliensis* [23]. Heterologous overexpression of the *RsICE1* gene improves the viability of transgenic rice under cold stress. The transcription levels of cold regulation genes *OsDREBl* and *OsTPP1* of transgenic rice were significantly up-regulated, increasing the soluble sugar content, chlorophyll content, and free proline content and reducing EL and MDA content [24]. Therefore, the study of ICE genes is a key link in studying the molecular mechanism of stresses.

*Malus baccata* is an important germplasm resource of apple resistance; it is widely distributed in the northeast, north, northwest and eastern Inner Mongolia areas of China. It has strong drought and cold resistance—some types can resist −50 °C—and contains valuable drought and cold resistance genes, which is of great significance for the stress-resistance breeding of apples. Research on model plant *Arabidopsis thaliana* shows that *AtICE1* and *AtICE2* can improve the cold tolerance of plants [25]. However, the research work on *MbICE1* has not yet begun. In this research, we demonstrated that transcription factor *MbICE1* functions to improve the resistance to cold and drought in *Arabidopsis* seedlings. Phylogenetic and structural analysis found that *MbICE1* belongs to a typical ICE family protein. To explore how *MbICE1* affects plant resistance, we analyzed the essential physiological indicators and crucial genes involved in cold and salt tolerance processes in control and transgenic seedlings. Under drought and cold, the overexpression of *MbICE1* enhanced antioxidant enzyme activities and the content of chlorophyll and proline, while inhibiting the accumulation of content of malondialdehyde (MDA), H_2_O_2_, and O2^−^ as well as inducing the up-regulation expression of *AtCBF* and its downstream function gene, resulting in the alleviation of damage from drought and cold to plants.

## 2. Results

### 2.1. Cloning and Bioinformatics Analysis of MbICE1

According to the MbICE1 protein sequence translated by DNAMAN5.2, its open reading frame (ORF) contains 1626 bp and encodes 541 amino acids. It was predicted that the theoretical isoelectric point (pI) of the MbICE1 protein was 5.81 and the theoretical molecular weight (MW) was 58.539 kDa. The predicted molecular formula was C_2510_H_4006_N_746_O_819_S_25_, average hydrophilic coefficient was −0.532, instability index was 48.54, and aliphatic index was 72.44_._ The contents of Leu (10.7%), Ser (10.5%), Gly (10.5%), Asn (8.3%), and Asp (7.0%) were higher in the MbICE1 protein. Therefore, we speculated that the MbICE1 protein was an unstable acidic hydrophilic protein. Sequence analysis showed that the sequence contained a conserved bHLH sequence composed of 50 amino acids between 349 aa and 398 aa. It can be seen that *MbICE1* belongs to the bHLH family (Appendix A).

### 2.2. Phylogenetic Relationship of MbICE1

In order to analyze the evolutionary relationship between ICE proteins in different plants, multiple-sequence alignment of MbICE1 with ICE proteins from 18 other species was conducted using DNAMAN5.2. MbICE1 protein contained a conserved serine-rich (S-rich) domain and ACT-like domain at the N-terminal and C-terminal, respectively, as well as conserved bHLH and NLS domains, which were consistent with other members of the ICE family. The sequence homology of the MbICE1 protein with *Arabidopsis* (AtICE1, AtICE2) was 60.7% and 54.5%, respectively, and 96.9% with *Malus domestica* (MdICE1) (Figure 1A). In addition, MbICE1 had the closest genetic relationship with MdICE1 and was relatively distant from AtICE1 and AtICE2 (Figure 1B). The TMDs of the MbICE1 protein was predicted by TMpred software. The MbICE1 protein had two transmembrane helices from the inner membrane to outer membrane, of 140 aa–159 aa and 491 aa–510 aa. There were two transmembrane helices from the outer membrane to inner membrane, of 151 aa–169 aa and 492 aa–511 aa (Appendix A).

The secondary structure of MbICE1 consists of 25.88%, 2.03%, 63.77%, and 8.32% α helix, β turn, random coils, and extended chain, respectively (Figure 2A). Using the SMART program to analyze the functional domain of the MbICE1 protein, it was found that an HLH motif was located on the MbICE1 protein with an E-value of 1.97e^−13^ (Figure 2B). In addition, the MbICE1 protein also contained AHS1, Sds3, HRDC, and other structures, which ensured that the MbICE1 protein could play a role in abiotic stress. The tertiary structure of the MbICE1 protein was predicted by Expasy online software, which was found to be consistent with the prediction of the secondary structure (Figure 2C).

### 2.3. Subcellular Localization of MbICE1 Protein

The fusion vector MbICE1-GFP was constructed and transformed into onion epidermal cells to determine the location of the MbICE1 protein. Evident green fluorescence was observed only in the nucleus (Figure 3E), while the whole cell showed green fluorescence in the positive control (35S:GFP) (Figure 3B). These results combined with the DAPI (4′, 6-diamidino-2-phenylindole) staining image (Figure 3F) demonstrated that MbICE1 was a nuclear localization protein.

### 2.4. Expression Analysis of MbICE1 in Malus Baccata

The mRNA expression level of *MbICE1* was analyzed by real-time fluorescence quantitative PCR (qRT-PCR). As shown in Figure 4A, *MbICE1* can be expressed in various tissues. The expression abundance was the highest in the stem, which was about 5.75 times that of the mature leaf. Under the four stresses, the expression of *MbICE1* in the new leaf and root increased first and then decreased with the duration of time. In the new leaf, the expression of *MbICE1* reached respectively the highest at 3 h, 6 h, 9 h, and 3 h under cold, drought, salt, and heat treatments, which were 11.74, 9.4, 6.79 and 5.52 times that of the untreated sample (CK) (Figure 4B). The expression of *MbICE1* in the root reached the highest at 9 h, 3 h, 9 h, and 6 h of cold, drought, salt, and heat treatments, at 10.51, 9.6, 6.27, and 4.89 times that of CK (Figure 4C). These data indicated that *MbICE1* expressions could be induced under all four stress treatments, while the up-regulated expression effects of drought and cold stresses were more obvious in *Malus baccata*.

### 2.5. Heterologous Expression of MbICE1 in Arabidopsis Improved Drought Tolerance

In order to determine whether the expression of *MbICE1* would affect the adaptive ability of plants in response to drought stress and cold stress, the pCAMBIA2300-MbICE1 vector was constructed, and the *MbICE1*-overexpressed *Arabidopsis* strains were generated under the control of the CaMV35S promoter. Six *MbICE1-*overexpressed positive strains (S1, S3, S4, S6, S8, and S9) after three successive generations of screening were identified by PCR gel electrophoresis among 10 speculated strains screened via kanamycin resistance (Figure 5A). Three homozygous positive lines, S1, S8, and S9, were randomly selected for further research with wild type (WT) and empty vector (VL) transformed *Arabidopsis* as the control.

To further determine the function of *MbICE1* under drought stress, a waterless stress assay was carried out for 7 days to analyze the phenotype and survival rate of *Arabidopsis*. There were no significant phenotype differences among the *Arabidopsis* lines (WT, VL, S1, S8 and S9) under control conditions (0 d drought). After drought stress, WT and VL lines suffered more obvious damage than transgenic *Arabidopsis* strains (S1, S8, and S9). After returning to normal conditions for 7 days, WT and VL seedlings almost died, while transgenic *Arabidopsis* strains could resume growth (Figure 5B). After returning to normal conditions for 7 days, the survival rates of all *Arabidopsis* lines were statistically analyzed. Under the control (0 d drought), all *Arabidopsis* lines basically survived. Specifically, when *Arabidopsis* was not watered for 7 days, 35.7% and 38.3% of WT and VL *Arabidopsis* survived, while 81.1%, 84.5%, and 78.3% of *MbICE1-*overexpressed *Arabidopsis* survived. Moreover, the water loss rates of *MbICE1-*overexpressed plant leaves were lower compared with the leaves of WT and VL, reflecting that overexpression of *MbICE1* was conducive to improving the water-holding capacity of *Arabidopsis* plants (Appendix A).

Furthermore, physiological indicators associated with drought response were determined. Under the control (0 d drought), there were no significant differences of the tested indicators among WT, VL, and *MbICE-* overexpressed *Arabidopsis* strains. Compared with normal conditions, the contents of malondialdehyde (MDA) and proline as well as the electrolyte leakage (EL) ratio of all *Arabidopsis* lines increased significantly after drought stress; however, the MDA content and EL of transgenic lines exhibited prominently lower levels than those of WT and VL lines, while the proline content showed the opposite trend (Figure 6A–C). Drought destroyed the chlorophyll structure so that its content decreased. In the present study, the chlorophyll content in *MbICE1-*overexpressed *Arabidopsis* was higher than that in control (Figure 6D). In keeping with the change in chlorophyll, the overexpression of *MbICE1* also enhanced the antioxidant enzyme activities, including superoxide dismutase (SOD), peroxidase enzyme (POD), and catalase enzyme (CAT) (Figure 6E–G). Additionally, drought led to a large accumulation of H_2_O_2_ and O2^−^ in plants, but the ectopic expression of *MbICE1* in *Arabidopsis* reduced this accumulation to some extent (Figure 6H,I). These data confirmed that the expression of *MbICE1* improved the drought tolerance of *Arabidopsis*.

### 2.6. Heterologous Expression of MbICE1 Improved Cold Tolerance in Arabidopsis

To further explore the function of *MbICE1* during stress, low temperature stress treatment was performed to analyze the phenotype and survival rate of *Arabidopsis*. Under cold stress, all tested *Arabidopsis* strains were damaged; however, the damage degrees of WT and VL strains were higher than transgenic lines. After returning to normal growth conditions, most WT and VL *Arabidopsis* died, while transgenic *Arabidopsis* gradually resumed growth (Figure 7A). Meantime, the survival rates of all *Arabidopsis* lines were statistically analyzed. When *Arabidopsis* strains exposed to −4 °C for 12 h were recovered in normal growth conditions for 7 days, 31.7% and 23.3% of WT and VL *Arabidopsis* survived, while 75.83%, 74.17%, and 71.67% of *MbICE1*-overexpressed *Arabidopsis* survived (Figure 7B). These results indicated that the overexpression of *MbICE1* was conducive to improving the adaptability of plants to cold stress.

To further determine the role of *MbICE1* in cold stress response, physiological indicators involved in cold response were analyzed. As shown in Figure 8, cold stress induced an increase of proline content, MDA content, and EL in *Arabidopsis*, but a decrease of chlorophyll content. After cold stress, compared with WT and VL lines, overexpression-*MbICE1 Arabidopsis* had lower MDA content and EL but higher proline and chlorophyll contents (Figure 8A–D). It was found that the overexpression of *MbICE1* could affect the antioxidant metabolism of *Arabidopsis* under cold stress. Cold stress increased the activities of POD, SOD, and CAT as well as the contents of H_2_O_2_ and O2^−^ in *Arabidopsis*. However, the antioxidant enzyme activities of overexpression-*MbICE1 Arabidopsis* increased more significantly, so its own ROS content accumulated less (Figure 8E–I). These results showed that overexpression-*MbICE1* can improve the cold resistance of *Arabidopsis*.

### 2.7. Stress Response Genes Were Up-Regulated by Overexpression of MbICE1 in Arabidopsis

The expressions of stress-related genes of overexpression-*MbICE1* plants under drought and cold stresses were analyzed by qRT-PCR. Under normal conditions, the expression of these genes in WT, VL, and transgenic plants was relatively lower. After stresses, the expressions of *AtCBF1*, *AtCBF*2, *AtCBF*3, *AtCOR15a*, *AtCOR47,* and *AtKIN*1 were significantly up-regulated in *MbICE1* transgenic lines (Figure 9). The above results showed that *MbICE1* can positively regulate the expression of the CBF gene and its downstream target genes, so as to improve the drought and cold resistances of transgenic *Arabidopsis.*

## 3. Discussion

Accumulating research has shown that overexpression of ICE genes can improve the viability of plants under adverse conditions [26]. For example, heterologous expression of *CsICE1*, an ICE transcription factor (TF) from tea tree, enhanced cold stress tolerance in *Arabidopsis.* Further studies showed that *CsICE1* promoted the accumulation of polyamines through the interaction with arginine decarboxylase ADC and played a positive role in the cold resistance of tea tree [19]. However, little research on ICEs TF has been conducted in *Malus baccata*. In this research, *MbICE1* was obtained by PCR amplification from *Malus baccata*. By analyzing the sequence homology, *MbICE1* was found to be a member of the ICE family, and its protein had typical structural features of the ICE family. *MbICE1* is a hydrophilic protein encoding 541 amino acids (Appendix A). The research found that Leu, Ser, and Gly contents in the ICE1 protein were the highest and predicted that the pI of the ICE1 gene in *M. domestica*, ‘*Muscat Hamburg*’, and Chinese cabbage was between 5.01 and 6.70 [20,27,28]. This study found that the PI of MbICE1 protein was 5.81, among which Leu, Ser, Gly, Asn, and Asp were the most abundant amino acids (Appendix A). The results were consistent with the previous predictions.

Through multiple-sequence alignment, it was found that the MbICE1 protein contained the typical bHLH domain, S-rich region, NLS domain, and ACT-like domain, which conformed to the structural characteristics of the ICE protein (Figure 1A). For example, the bHLH domain can recognize and combine MYC sites in the CBF promoter, thereby changing the expression of target genes [29]. Dicotyledon ICE proteins contain S-rich regions, such as AtICE1 and VaICE1, while many monocotyledons do not contain typical S-rich regions, such as OsICE1 and ZmmICE1 [14,30,31]. The MbICE1 protein had a hypothetical NLS region, indicating that the MbICE1 protein may act as a transcriptional activator [28], which can regulate the expression of stress-related genes. It can be seen from the evolutionary tree that the MbICE1 protein was closely related to *M. domestica* (Figure 1B).

In this study, the 35S:MbICE1-GFP fusion vector was transferred into onion epidermal cells by gene gun bombardment method. It was found that the target gene was localized on the nucleus under a confocal microscope. This result was consistent with previous studies, which also proved that MbICE1, like HbICE2, CfICE1, ZjICE1, and CmICE2 [22,23,32,33], mainly plays a role in the nucleus (Figure 3).

The ICE gene is expressed in different tissues of the plant. *AtICE1* can be expressed in roots, stems, leaves, and flowers of *Arabidopsis* [34]. The expression of *HbICE1* was higher in the stem, leaf, and flower, but lower in the root [35]. In this study, *MbICE1* can be expressed in the new leaf, stem, root, and mature leaf; levels were the highest in the stem and the lowest in the mature leaf (Figure 4A). In summary, we found that the expression of ICE transcription factors in different parts of the same plant was different, and the expression patterns in different plants were similar. Studies have found that the expression of ICE is regulated by stresses. Cold, drought, high salt, and oxidative stresses positively regulated the expression of *SlICE1a* [36]. The expression of *LsICE1*, *BcICE1,* and *RsICE1* was induced by cold and saliferous conditions, but they were not sensitive to drought stress [28]. These studies showed that ICE genes not only participated in the response pathway of cold stress but also affected the growth of plants under high salt, drought, and oxidative stresses. In this study, the results showed that *MbICE1* was induced by abiotic stresses such as drought, cold, salt, and heat treatments. The expression of *MbICE1* in new leaves and roots was more sensitive to drought and cold stresses (Figure 4B,C).

Many genes have been verified in model plants for their functions [21,22,23]. The overexpression of *AnICE1* in *Arabidopsis* improves the cold tolerance of plants [37]. In order to further verify the function of *AnICE1*, an overexpression vector was transferred into *Piptanthus mongolicus* seedlings. The results showed that the physiological and biochemical results of transgenic *Piptanthus mongolicus* plants under cold stress corresponded to those of transgenic *Arabidopsis* plants [37]. Because transgenic apple seedlings are difficult to obtain, we can first analyze their functions under drought and cold stresses by overexpressing the *MbICE1* gene in *Arabidopsis.* The overexpression vector pCAMBIA2300-MbICE1 was transferred into *Arabidopsis*. Because drought and low temperature treatments could significantly induce the upregulation of *MbICE1* expression, these two kinds of stresses were used for the next functional research. All tested plants were damaged to a certain extent, but the damage degree of *MbICE1* transgenic *Arabidopsis* was lesser than that of WT and VL plants, and the survival rate was significantly higher than that of WT and VL plants under drought and cold stresses. It was speculated that *MbICE1* improved the tolerance of transgenic plants to drought and cold stresses (Figure 5 and Figure 7). The EL and MDA content were indicators to judge whether plant cells were damaged [38,39,40]. Overexpression of *VaICE1* resulted in the EL and MDA content of transgenic plants being less than WT [41]. Similarly, overexpression of *SmICE1a* reduced the electrolyte leakage of transgenic *Arabidopsis* and improved the cold resistance [13]. Drought and cold stresses caused changes in chlorophyll, MDA, proline, and electrolyte leakage of plants, which was related to the expression of stress response genes. The chlorophyll contents of WT, VL, and transgenic plants decreased after stress treatments, but that of transgenic plants was usually higher. Stresses reduce plant metabolism, affects the biosynthesis of photosynthetic pigments, and affects chlorophyll content and photosynthetic capacity [42]. The decrease of chlorophyll content under drought and cold conditions may mean that various abiotic stresses enable plants to start their defense systems. This defense response was conducive to improving the survival rate of plants under stress.

In this study, overexpression-*MbICE1* transgenic lines had stronger viability under stress, which was similar to the overexpression of *MdICE1* in *Arabidopsis*. In addition, after stress treatment, the EL and MDA content of WT and VL plants increased while the proline content was lower than that of overexpression-*MbICE1 Arabidopsis*, which was consistent with the higher tolerance of the overexpression-*MbICE1* line. Abiotic stress induced the accumulation of H_2_O_2_ and O^2−^ [43,44,45,46], resulting in the lipid peroxidation of biofilm. When plants are in drought and cold environment, ROS content accumulates greatly and antioxidant enzyme activities also increase. Overexpression of *MbICE1* can improve the activities of POD, SOD, and CAT; the increase of antioxidant enzyme activities enhanced the scavenging capacity of *Arabidopsis* to ROS, reduced the peroxidation of ROS on the cell plasma membrane, and enhanced the adaptability of *Arabidopsis* to drought and cold (Figure 6 and Figure 8).

The ICE transcription factor is the most upstream gene known in the ICE-CBF-COR expression cascade [8]. Under stresses, ICE TF can recognize MYC elements of CBF to initiate the ICE-CBF-COR pathway. Therefore, the role of ICE transcription factor in abiotic stress regulation cannot be ignored. Drought and cold stresses significantly induced the expression of *AtCBF1*, *AtCBF2*, *AtCBF3*, *AtCOR15a*, *AtCOR47,* and *AtKIN1* in *Arabidopsis* (Figure 9). In addition, due to the differences in the expression of different CBFS, we speculate this may be related to the different regulatory pathways of ICE. Therefore, it is speculated that *MbICE1* positively regulated the expression of CBF and its downstream target genes through differential expression, so as to improve the drought and cold resistance of plants.

*MbICE1* of *Malus baccata* was isolated and characterized. When *MbICE1* receives the signal of drought and cold, it directly combines with the MYC (CANNTG) element in the CBF promoter to activate the expression of *AtCBF1*, *AtCBF2,* and *AtCBF3*, thus positively regulating the expression of *AtCOR15a* and *AtCOR47* and improving the survival ability of *Arabidopsis* under drought and cold stresses. At the same time, we speculated that the accumulation of *MbICE1* may regulate the ABA signal pathway by activating the RING-type E3 ubiquitin ligase *AtPPRT3*, promoting the stomatal closure of *Arabidopsis* and inhibiting water loss [47]. In addition, the accumulation of ABA increased the expression of *AtKIN1*, thus enhancing the adaptation of plants to drought and cold (Figure 10). In general, overexpression of *MbICE1* was conducive to improving plant survival under drought and cold stresses.

## 4. Materials and Methods

### 4.1. Plant Materials, Growth Conditions, and Treatments

The seedlings of *Malus baccata* were planted according to the method of Han et al. [48]. The *Malus baccata* seedlings with 10–12 leaves and well-developed roots were treated with cold (4 °C), salt (200 mM NaCl), drought (20% PEG6000), and heat (37 °C) stresses [49]. Then, samples of different treatments were immediately collected at 0 h, 1 h, 3 h, 6 h, 9 h, and 12 h. In addition, the expression of *MbICE1* was analyzed by qRT-PCR in different tissues of normal (Control) growing hydroponic seedlings. The Columbia wild-type (WT) and *MbICE1-*overexpressed *Arabidopsis* seedling were planted in an illumination incubator, and *Arabidopsis* plants experienced light and darkness in a ratio of 2:1 over a 24 h period [50].

### 4.2. Cloning and Bioinformatic Expression Analysis of MbICE1

Total RNA was extracted from *Malus baccata* and reverse transcribed into the first strand of cDNA. Cloning primers (*MbICE1*-F/R: 5′-ATGCTGCCAAGGCTGAACGGT-3′/5′-CTACAT CATGCCATGGAACCCGA-3′) were designed using the CDS region of *MdICE1* (MD14G1148600, *M. domestica*) as the reference sequence. The method of Han et al. [51] was used to obtain escherichia coli containing *MbICE1*. DNAMAN5.2, NCBI, MEGA 5.0, and online software Expasy were used to predict the homology and physicochemical properties of MbICE1 protein [52,53].

### 4.3. Subcellular Localization of MbICE1

The coding sequence (CDS) removing the termination codon of *MbICE1* was cloned with primers 5′-cgGGATCCATGCTGCCAAGG-3′ (*BamH* I) and 5′-gcGTCGACCATCATGCCAT G-3′ (*Sal* I) and inserted into the pSAT6-GFP-N1 vector, forming fusion expression vector 35S:MbICE1-GFP. Subsequently 35S:MbICE1-GFP and empty vectors (35s:GFP as a positive control) were transformed into onion epidermal cells by gene gun bombardment [54]. 4′, 6-diamino-2-phenylindole (DAPI) reagent is a fluorescent dye that can strongly bind to DNA and is commonly used for nuclear detection and localization. A confocal microscope (LSM 900, Precise, Beijing, China) was used to record the location of 35S:MbICE1-GFP and control protein fluorescence [55].

### 4.4. Quantitative Real-Time PCR (qRT-PCR) Analysis of MbICE1

The primers *MbICE1*-qF and *MbICE1*-qR (*MbICE1*-qF/R: 5′-AAGGCTGAACGGTGG TG-3′/5′-CCTTGTTCTCGGTGTTGC-3′) were designed for real-time PCR [56]. The primers *MbActF1*/*R1* (5′-ACACGGGGAGGTAGTGACAA-3′/5′-CCTCCAATGGATCCTCGTTA-3′) and *MbUBQF1*/*R1* (5′-AAATCCAGGACAAGGAGGGC-3′/5′-CACCACGGAGACGCAAC AC-3′) were planned based on the *Malus Actin* gene (EB127077) and *UBQ* (U74358.1) as the internal reference control of qRT-PCR. The method of Liang et al. [44] was used for qRT-PCR operation. The expression level of *MbICE1* was analyzed by the 2^−ΔΔCT^ method [57].

### 4.5. Vector Construction and Generation of Transgenic Lines

To construct the pCAMBIA2300-MbICE1 overexpression vector, an *MbICE1* target fragment with *Sal*I and *BamH*I restriction sites (recombinant primer: 5′-GGACAGGGTACCCGGG GATCCATGCTGCCAAGGCTG-3′/5′-CTGGCATGCCTGCAGGTCGACCATCATGCCAT GGAA-3′) was ligated into the pCAMBIA2300 vector using homologous recombinase (Novizan C115-01/02). Then, the agrobacterium tumefaciens GV3101 with resulting vector pCAMBIA2300-MbICE1 and empty pCAMBIA2300 were transfected into the wild-type *Arabidopsis* seedlings by floral dip transformation [58,59]. Transgenic strains and vector line (VL, only transformed with pCAMBIA2300 vector) were planted on the screening medium containing resistance (50 mg L^−1^ kanamycin) for further analysis. Positive transgenic plants were confirmed by PCR amplification of the *MbICE1*. T_3_ homozygous *A. thaliana* seedlings displaying 100% kanamycin resistance were used for further research.

### 4.6. Physiological Measurements

The water loss rates of WT, VL, and transgenic seedlings were measured [60] and subjected to stress treatments for 25 days. The WT, VL, and overexpression-*MbICE1* lines (S1, S8, and S9) of *Arabidopsis* were treated with drought (not watered for 7 days) and cold (−4 °C for 12 h) stresses, and the control group (25 °C) was simultaneously established. Then, the *Arabidopsis* was watered again under 25 °C conditions. Survival was assessed after 7 days.

The contents of chlorophyll, malondialdehyde (MDA), free proline (Pro), and reactive oxygen species (ROS: H_2_O_2_ and O^2−^) and the activities of SOD, POD, and CAT were determined by assay kit (Suzhou Comin Biotechnology Co., China). The electrolyte leakage (EL) of the sample was determined by Campos et al. [39]. The measurements were executed three times.

### 4.7. Analysis of Downstream Gene Expression of MbICE1

To collect the RNA of leaf samples under the drought and cold conditions described above, qPCR was used to analyze the expression of stress-related genes (*AtCBF1*, *AtCBF2*, *AtCBF3*, *AtCOR47*, *AtCOR15a,* and *AtKIN1*) in transgenic, VL, and WT *Arabidopsis*. *AtACT2* and *AtUBQ10* were used as internal parameters. The specific primers were as follows: *AtCBF1*-F/R: 5′-GAGACTCGTCACCCAATTTACAGA-3′/5′-CTGCCATCTCAGCGGTTTG-3′; *AtCBF2*-F/R: 5′-GACCTTGGTGGAGGCTATTT-3′/5′-ATCCCTTCGGCCATGTTATC-3′; *AtCBF3*-F/R: 5′-TTTCGCTGACTCGGCTTGG-3′/5′-CCGCCTTTTGGATGTCCTTAG-3′; *AtCOR47*-F/R: 5′-GGCTGAGGAGTACAAGAACAA-3′/5′-ACAATCCACGATCCGTAACC-3′; *AtCOR15a*-F/R: 5′-GGCGTATGTGGAGGAGAAAG-3′/5′-CCCTACTTTGTGGCATCCTT AG-3′; *AtKIN1*-F/R: 5′-GCAATGTTCTGCTGGACAAG-3′/5′-TCCTTCACGAAGTT AACACCTC-3′; *AtACT2*-F/R: 5′-TTACCCGATGGGCAAGTC A-3′/5′-AAACGAGGGCTG GAACAAGA-3′; *AtUBQ10*-F/R: 5′-CACACTCCACTTGGTCTTGC GT-3′/5′-TGGTCTTTC CGGTGAGAGTCTTC A-3′ [61].

### 4.8. Statistical Analysis

GraphPad prism (v8.0.2.263) software was used to analyze the data of physiological indexes and mark the standard errors (SE). All biological and operational measurements were repeated three times. The differences between the data were marked as significant (* *p* ≤ 0.05, ** *p* ≤ 0.01).

## 5. Conclusions

In this study, a new ICE gene was cloned from *Malus baccata* and named *MbICE1*. According to the fluorescence signal observed by the confocal microscope, MbICE1 is a nuclear protein. In addition, drought and cold induced high expression of *MbICE1* in new leaves and roots. The overexpression of *MbICE1* in *Arabidopsis thaliana* can significantly improve the drought and cold tolerance of plants by regulating the expression of CBF (*AtCBF1*, *AtCBF2,* and *AtCBF3*) and its downstream target genes (*AtCOR47*, *AtCOR15a,* and *AtKIN1*), enhancing the antioxidant enzyme system, and maintaining the ROS stability and membrane lipid stability under stress conditions. Taken together, our results indicate that *MbICE1* plays an active role in response to drought and cold stresses. In the following experiments, we will aim to transfer the *MbICE1* overexpression vector into apple *Gl-3* to further verify its function.

## Figures and Tables

**Figure 1 ijms-23-16072-f001:**
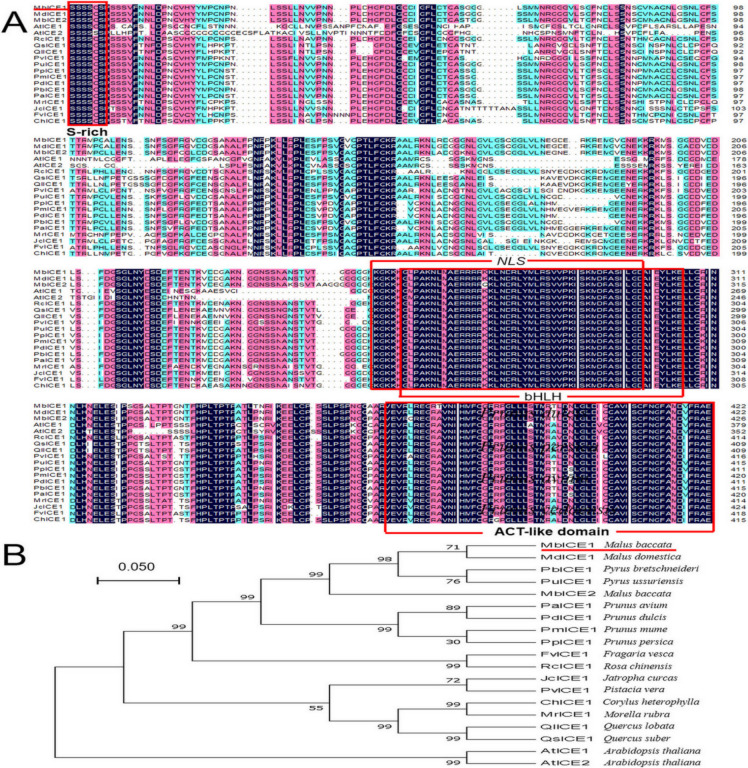
Multi-sequence alignment and evolutionary relationship analysis of MbICE1 protein. (**A**) Multiple sequence alignment. (**B**) Evolution tree. The target protein is indicated with a red line. GenBank ID: MdICE1 (NP_001280967.1), PbICE1 (XP_009337514.1), PuICE1 (APC57593.1), PaICE1 (XP_021814716.1), PdICE1 (XP_034216895.1), PmICE1 (XP_008239552.1), PpICE1 (XP_007209976.2), FvICE1 (XP_004297495.2), RcICE1 (XP_024168837.1), JcICE1 (NP_001306859.1), PvICE1 (XP_031277289.1), ChICE1 (ADZ48234.1), MrICE1 (KAB1217526.1), QlICE1 (XP_030939307.1), QsICE1 (XP_023877763.1), AtICE1 (NP_001030774.1), AtICE2 (NP_172746.2).

**Figure 2 ijms-23-16072-f002:**
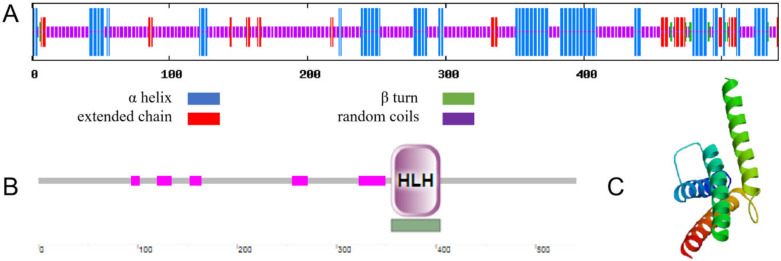
Structure and domain prediction of MbICE1. (**A**) Secondary structure of MbICE1. (**B**) Functional domain of MbICE1. (**C**) Tertiary structure of MbICE1.

**Figure 3 ijms-23-16072-f003:**
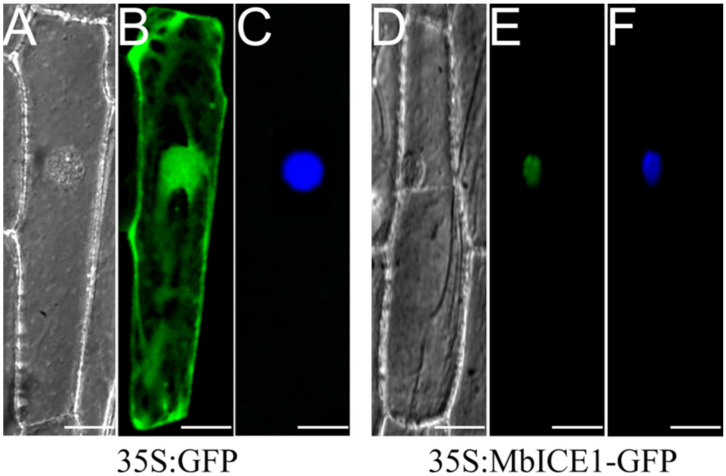
Subcellular localization of MbICE1. The transient vector harboring 35S:GFP and 35S:MbICE1-GFP vectors were transformed into onion epidermal cells by gene gun bombardment. (**A**,**D**) Bright field of vision. (**B**,**E**) GFP signals in dark field. (**C**,**F**) Effects after DAPI dyeing. The white scale bar represents 5 μm.

**Figure 4 ijms-23-16072-f004:**
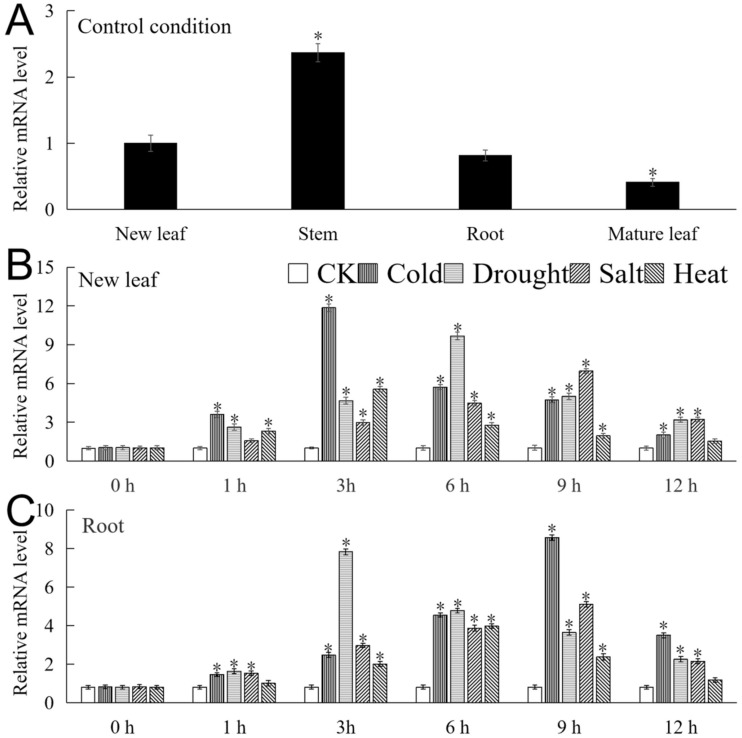
The expression of *MbICE1*. (**A**) Expression of *MbICE1* gene in different tissues of *Malus baccata* under normal growing conditions. The new leaf was selected as the control group. (**B**) The relative expression of *MbICE1* gene in *Malus baccata* new leaf and (**C**) root under control conditions (CK, no stress treatment was performed) and stress conditions. The value is the average of three repeats, and the standard deviation is represented by the error bar. The asterisk indicates the difference between treatment and control conditions (* *p* ≤ 0.05).

**Figure 5 ijms-23-16072-f005:**
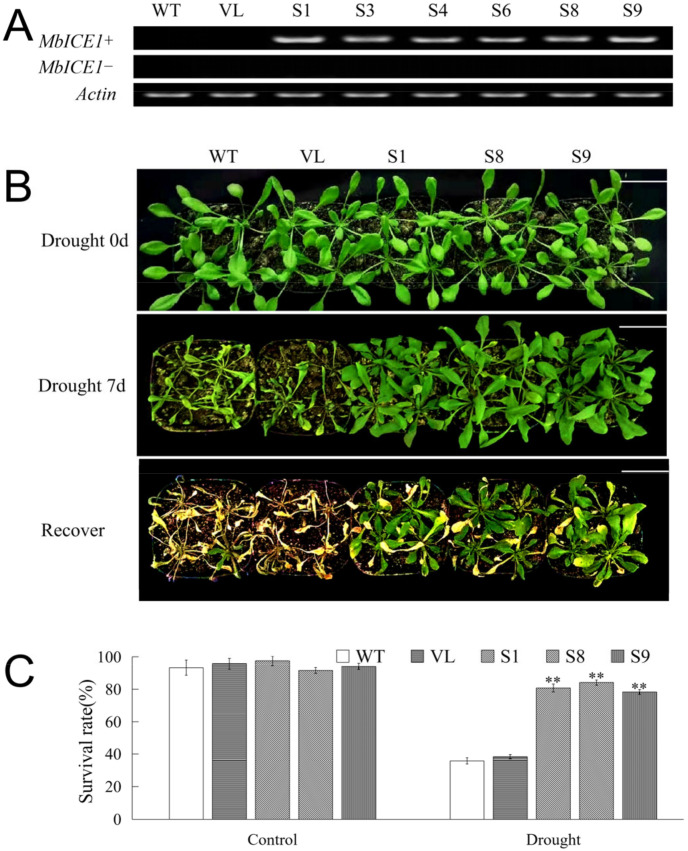
Overexpression of *MbICE1* improves drought resistance of plants. (**A**) PCR was used to verify the expression of *MbICE1* in WT, VL, and transgenic *Arabidopsis*. The actin gene (AB638619.1) was selected as the control group. (**B**) Phenotypes of WT, VL (vector line), and overexpression-*MbICE1* (S1, S8, and S9) under control, drought, and recovery growth. The white scale bar represents 3 cm. (**C**) Survival rates of WT and overexpression-*MbICE1* lines after Drought 0 d and Drought 7 d. The value is the average of three repeats. The standard deviation is represented by the error bar. The asterisk indicates the difference compared to WT (** *p* ≤ 0.01).

**Figure 6 ijms-23-16072-f006:**
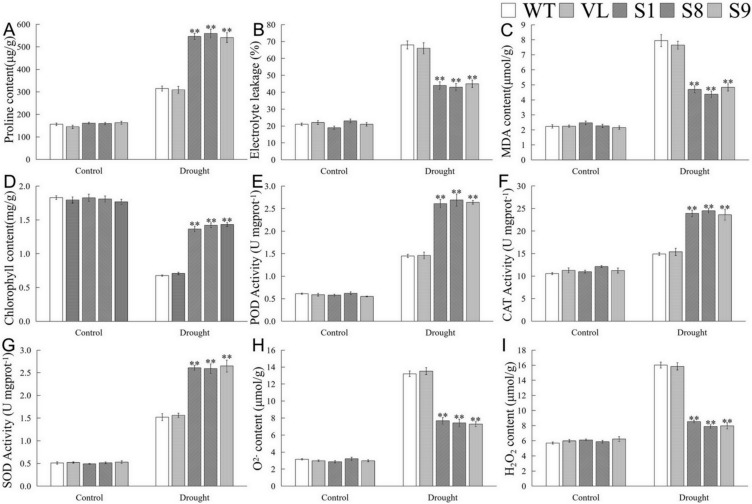
Evaluation of physiological indices responsive to drought stress. (**A**) Proline content (Pro); (**B**) electrolyte leakage (EL); (**C**) malondialdehyde content (MDA); (**D**) chlorophyll content; (**E**) peroxidase activity (POD); (**F**) catalase activity (CAT); (**G**) superoxide dismutase activity (SOD); (**H**) O2^−^ content; (**I**) H_2_O_2_ content in WT, VL (vector line), and overexpression-*MbICE1* under control and drought. The value is the average of three repeats, and the standard deviation is represented by the error bar. The asterisk indicates difference among WT, VL, and overexpression-*MbICE1* (** *p* ≤ 0.01). Each index of WT in each treatment was used as the control group.

**Figure 7 ijms-23-16072-f007:**
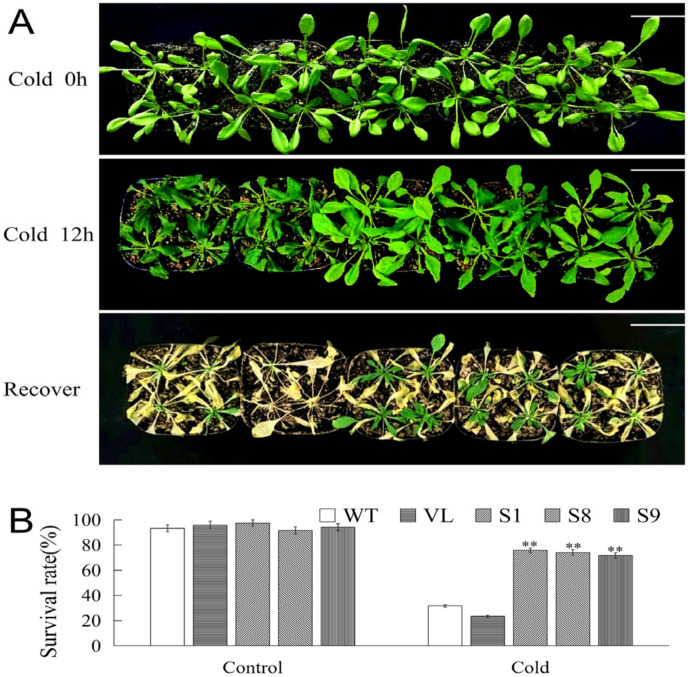
Overexpression of *MbICE1* improves cold resistance of plants. (**A**) Phenotypes of WT, VL, (vector line) and overexpression-*MbICE1* under Cold 0 h, Cold 12 h, and recover growth. The white scale represents 3 cm. (**B**) Survival rates of WT, VL, and overexpression-*MbICE1* after Cold 0 h and Cold 12 h. The value is the average of three repeats, and the standard deviation is represented by the error bar. The asterisk indicates the difference compared to WT (** *p* ≤ 0.01).

**Figure 8 ijms-23-16072-f008:**
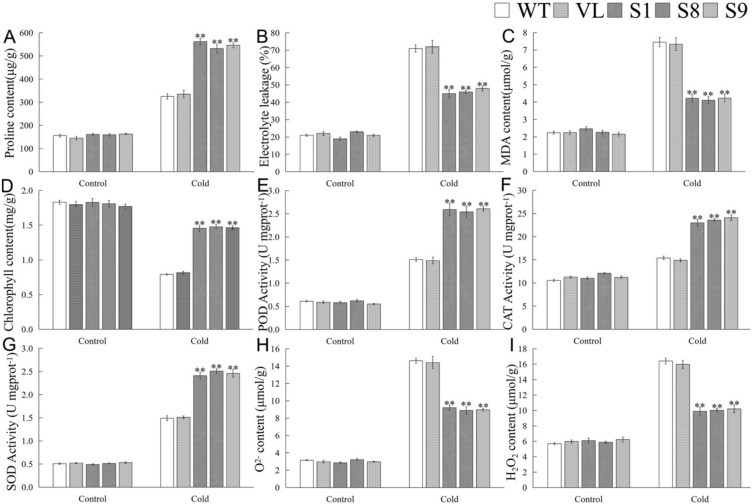
Evaluation of physiological indices responsive to cold stress. (**A**) Proline content (Pro); (**B**) electrolyte leakage (EL); (**C**) malondialdehyde content (MDA); (**D**) chlorophyll content; (**E**) peroxidase activity (POD); (**F**) catalase activity (CAT); (**G**) superoxide dismutase activity (SOD); (**H**) O2^−^ content; (**I**) H_2_O_2_ content in WT, VL (vector line), and overexpression-*MbICE1* under Cold 0 h and Cold 12 h. The value is the average of three repeats, and the standard deviation is represented by the error bar. The asterisk indicates difference among WT, VL, and overexpression-*MbICE1* (** *p* ≤ 0.01). Each index of WT in each treatment was used as the control group.

**Figure 9 ijms-23-16072-f009:**
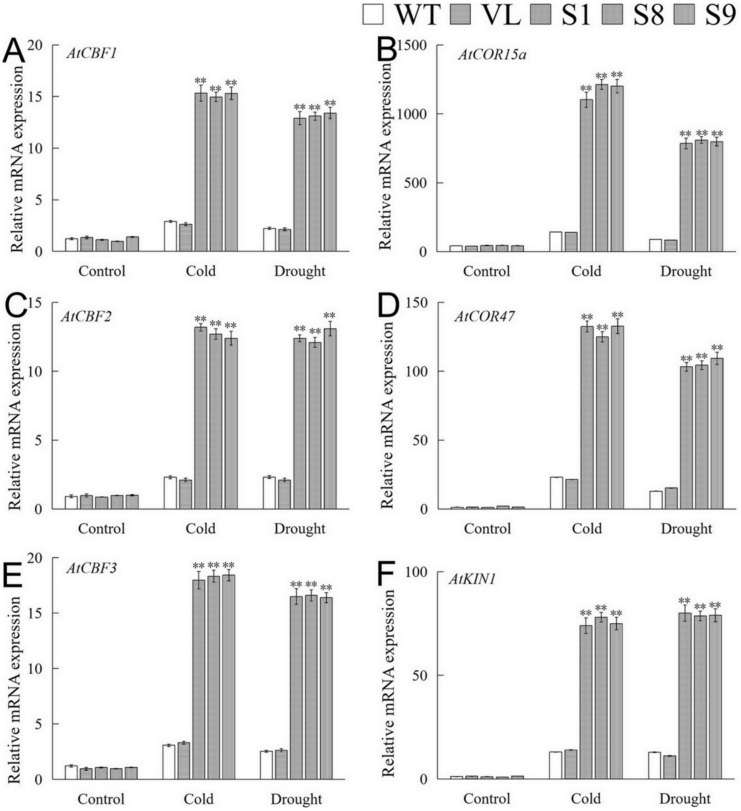
Analysis of downstream gene expression of *MbICE1* in WT, VL (vector line), and overexpression-*MbICE1* under drought and cold stresses. Relative expression level of (**A**) *AtCBF1*; (**B**) *AtCOR15a*; (**C**) *AtCBF2*; (**D**) *AtCOR47*; (**E**) *AtCBF3*; (**F**) *AtKIN1* by q-PCR. The value is the average of three repeats, and the standard deviation is represented by the error bar. The asterisk indicates the difference compared to WT (** *p* ≤ 0.01).

**Figure 10 ijms-23-16072-f010:**
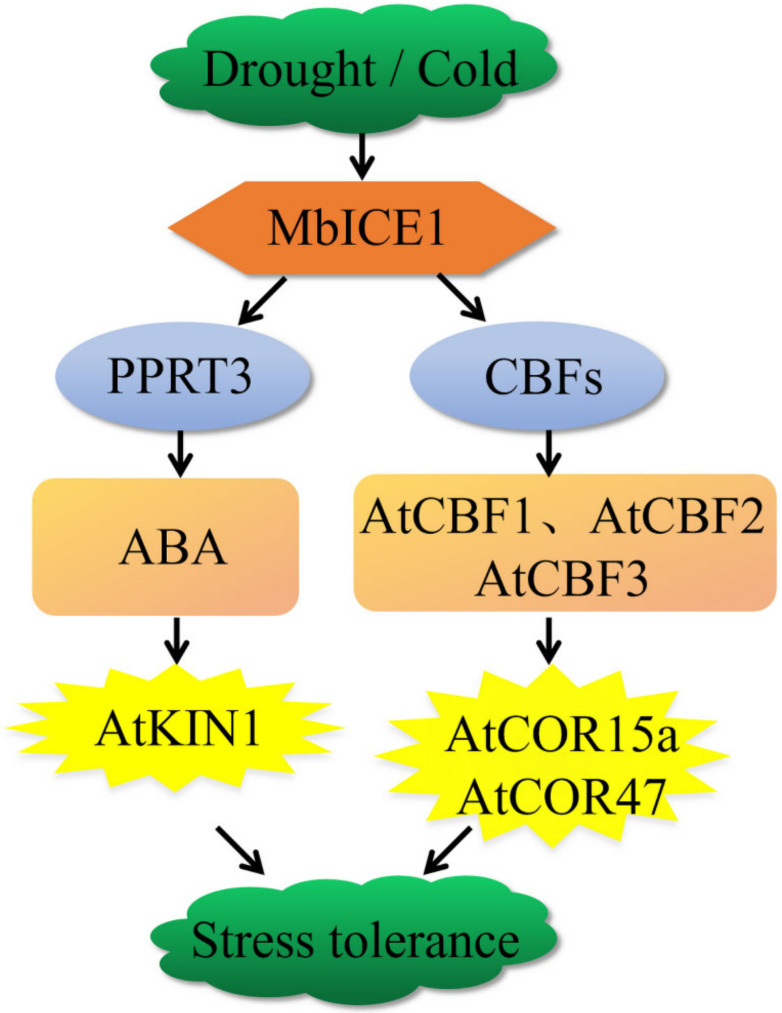
A model for drought or cold adaptation mediated by *MbICE1*. *MbICE1* receives the signals of drought and cold and directly combines with MYC elements in the CBF promoter to activate the expression of CBFs, thus positively regulating the expression of *AtCOR15a* and *AtCOR47* and improving the survival ability of *Arabidopsis* under drought and cold stress. In addition, the accumulation of *MbICE1* may regulate the ABA signal pathway by activating the RING type E3 ubiquitin ligase *AtPPRT3*, thus increasing the expression of *AtKIN1* and enhancing the plant’s adaptation to drought and cold.

## Data Availability

The original data in this study are available from the corresponding authors.

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
