# Peer review of "MbICE1 Confers Drought and Cold Tolerance through Up-Regulating Antioxidant Capacity and Stress-Resistant Genes in Arabidopsis thaliana"

_ijms, 2022, doi:10.3390/ijms232416072_

Round 1

Reviewer 1 Report

The manuscript titled (MbICE1 Confers Drought and Cold Tolerance through Up-regulating Antioxidant Capacity and Stress Resistant Gene in Arabidopsis thaliana( by Duan et al. Isolated MbICE1 gene from M. baccata. The function of MbICE1 under drought and cold stresses was analyzed by overexpression technology.
The manuscript is generally well-written and structured. The analysis was successful, and the data was well understood and modeled in detail. In addition, the manuscript contains relevant paragraphs that have been discussed. The selection of the bibliography is appropriate to the content of the manuscript.
However, some errors appeared throughout the manuscript, making it difficult to accept it in its current version.
-    In the introduction section, the authors have to explain the importance of  Malus baccata
-    All Gene names in the manuscript must be in italics.
-     MDA in the abstract must be (Malondialdehyde).
-    Why did the authors not do Multi-sequence alignment and evolutionary relationship analysis of MbICE1 using cDNA?
-    What is the meaning of CK in figure 4?
-    In lines 358 and 359, the authors must explain how they did the treatment of cold, salt, drought, and hot stresses and replace the hot word with the heat word.
-    According to the expression analysis results, cold, drought, salt, and heat can induce the up-regulated expression of MbICE1 in M.baccata. Why the authors tested overexpression-MbICE1 lines just under drought and cold? What about salinity and heat stress?
-    Write the meaning of VL in all the figure's legends.
-    Must write legends for the supplemental figures.
-    Lines 173 (all strawberry lines survived basically)????????
-    Line 279 (In this study, MbICE1-GFP fusion vector was transferred into onion epidermal cells by gene gun bombardment method) must be mentioned in the M&M section, and also in the figure 3 legend must mention the used material  (onion epidermal cells).
-    Line 296 replaces hot with heat.
Best Regards

Author Response

The manuscript titled (MbICE1 Confers Drought and Cold Tolerance through Up-regulating Antioxidant Capacity and Stress Resistant Gene in Arabidopsis thaliana( by Duan et al. Isolated MbICE1 gene from M. baccata. The function of MbICE1 under drought and cold stresses was analyzed by overexpression technology.

The manuscript is generally well-written and structured. The analysis was successful, and the data was well understood and modeled in detail. In addition, the manuscript contains relevant paragraphs that have been discussed. The selection of the bibliography is appropriate to the content of the manuscript.

However, some errors appeared throughout the manuscript, making it difficult to accept it in its current version.

We appreciate your valuable suggestions and advices on our manuscript. I think they are very helpful and important, and revisions had been made in the revised manuscript accordingly.

Here I would like to response the comments and add some explanations as follows. Supplementary materials and pictures have also been corrected.

In the introduction section, the authors have to explain the importance of Malus baccata.

Response: Yes, we think it's a good suggestion. we have accepted your suggestion and have introduced the importance of Malus baccata in the introduction.

All Gene names in the manuscript must be in italics.

Response: Yes, we have accepted your suggestion and we have corrected this part.

MDA in the abstract must be (Malondialdehyde).  

Response: Yes, we have accepted your suggestion and have replaced "MDA" with " Malondialdehyde".

Why did the authors not do Multi-sequence alignment and evolutionary relationship analysis of MbICE1 using cDNA?

Response: Yes, we think it's a good suggestion. However, we thought it would be better to use protein multi-sequence alignment in this paper. Protein multi-sequence alignment usually has more information than cDNA multi-sequence alignment. In addition, proteins are more conserved than cDNA sequences. Protein multi-sequence alignment is more conducive to the comparison between different species. Of course, we will also use cDNA multi-sequence alignment for analysis in future studies.

What is the meaning of CK in figure 4?

Response: Yes, we have accepted your suggestion and have explained the meaning of CK in the figure 4.

In lines 358 and 359, the authors must explain how they did the treatment of cold, salt, drought, and hot stresses and replace the hot word with the heat word.

Response: Yes, we have accepted your suggestion and have added the specific conditions of cold, salt, drought and heat treatments in the methods. And we have replaced "hot" with " heat".

According to the expression analysis results, cold, drought, salt, and heat can induce the up-regulated expression of MbICE1 in M.baccata. Why the authors tested overexpression-MbICE1 lines just under drought and cold? What about salinity and heat stress?

Response: Yes, we think it's a good suggestion. In this manuscripts, after the four stress treatments, it was found that compared with the CK, the expression of MbICE1 gene was more significantly different under cold and drought stresses, and MBICE1 gene was more sensitive to cold and drought. Therefore, cold and drought stresses were selected to treat the overexpression-MbICE1 lines. Of course, we will also treat overexpression-MbICE1 plants with salt and heat stresses in future studies.

Write the meaning of VL in all the figure's legends.

Response: Yes, we have accepted your suggestion and we have supplemented this part.

Must write legends for the supplemental figures.

Response: Yes, we have accepted your suggestion and we have supplemented this part.

Lines 173 (all strawberry lines survived basically)????????  

Response: Yes, we have accepted your suggestion. This is a mistake in our writing and we have corrected this part

Line 279 (In this study, MbICE1-GFP fusion vector was transferred into onion epidermal cells by gene gun bombardment method) must be mentioned in the M&M section, and also in the figure 3 legend must mention the used material  (onion epidermal cells).

Response: Yes, we have accepted your suggestion and have supplemented this part. 

Line 296 replaces hot with heat.

Response: Yes, we have accepted your suggestion and have replaced "hot" with " heat".

Reviewer 2 Report

Dear authors,

Manuscript ijms-2102631 entiteled "MbICE1 Confers Drought and Cold Tolerance through Up-regulating Antioxidant Capacity and Stress Resistant Gene in Arabidopsis thaliana" and authored by Yadong Duan , Jiaxin Han , Baitao Guo , Wenbo Zhao , Shuang Zhou , Chunwei Zhou , Lei Zhang , Xingguo Li and Deguo Han * fits well with the scope of the journal. The paper documents several evidences that support MbICE1 from the apple rootstock Malus baccata (L.) Borkh as a major actor  in plant drought and cold resistance. While the experiments have been nicely designed and well conducted few issues needs the attention of the authors before the acceptance of the manuscript:

1. Introduction section is very weak. There is no explanation for readers outside of the field about antioxydant metabolism. The precise omode of action of H2O2, MDA, antioxydant enzymes and their relationships with drought and cold stress.

2. Introduction: please explain the strategy you are using here : expressing a gene in Arabidopsis and trying to get responses in apple trees. Please just tell the readers about the success of this strategy in litterature and its limits mainly !

3. Results section: in their current form the figures 1,4, 6, 8 and 9 are not readable. Therefore I suggest improving the resolution of these figures. I am afaraid readers could not analyse them.

4. There is no discussion about the validity of the strategy used in this study. is improvement of drought and cold resistance in the weed Arabidopsis a proof that this gene have the same function than in the tree Malus baccata (L.) Borkh. The limits of your strategy have to be discussed or your results will have the state of speculation. How valid is this strategy. Have it been applied before in apple trees research or in similar species. Please discuss these issues.

5. There is no conclusion of this manuscript. Please dedicate a section to conclusions and outlook. In this part you need to present main findings of the manuscript, highlight their input to the field and the future needed experiments to go further in the field.

I am really waiting to read an improved version of this manuscript that I can recommand for publication.

Best regards

Author Response

Manuscript ijms-2102631 entiteled "MbICE1 Confers Drought and Cold Tolerance through Up-regulating Antioxidant Capacity and Stress Resistant Gene in Arabidopsis thaliana" and authored by Yadong Duan , Jiaxin Han , Baitao Guo , Wenbo Zhao , Shuang Zhou , Chunwei Zhou , Lei Zhang , Xingguo Li and Deguo Han * fits well with the scope of the journal. The paper documents several evidences that support MbICE1 from the apple rootstock Malus baccata (L.) Borkh as a major actor  in plant drought and cold resistance. While the experiments have been nicely designed and well conducted few issues needs the attention of the authors before the acceptance of the manuscript:

We appreciate your valuable suggestions and advices on our manuscript. I think they are very helpful and important, and revisions had been made in the revised manuscript accordingly.

Here I would like to response the comments and add some explanations as follows.

Supplementary materials and pictures have also been corrected.

  1. Introductionsection is very weak. There is no explanation for readers outside of the field about antioxydant metabolism. The precise omode of action of H2O2, MDA, antioxydant enzymes and their relationships with drought and cold stress.

Response: Yes, we have accepted your suggestion and we have supplemented this part.

  1. Introduction: please explain the strategy you are using here : expressing a gene in Arabidopsis and trying to get responses in apple trees. Please just tell the readers about the success of this strategy in litterature and its limits mainly !

Response: Yes, we think it's a good suggestion. And we added this part in the introduction.

  1. Resultssection: in their current form the figures 1,4, 6, 8 and 9 are not readable. Therefore I suggest improving the resolution of these figures. I am afaraid readers could not analyse them.

Response: Yes, we have accepted your suggestion and have replaced the photos with clearer images.

  1. There is no discussion about the validity of the strategy used in this study. is improvement of drought and cold resistance in the weed Arabidopsis a proof that this gene have the same function than in the tree Malus baccata (L.) Borkh. The limits of your strategy have to be discussed or your results will have the state of speculation. How valid is this strategy. Have it been applied before in apple trees research or in similar species. Please discuss these issues.

Response: Yes, we have accepted your suggestion and have supplemented this part  discussion.

  1. There is no conclusion of this manuscript. Please dedicate a section to conclusions and outlook. In this part you need to present main findings of the manuscript, highlight their input to the field and the future needed experiments to go further in the field.

Response: Yes, We have accepted your suggestion and we have supplemented this part.

Reviewer 3 Report

Dear Authors,

I have an honor to review manuscript entitled “MbICE1 Confers Drought and Cold Tolerance through Up-regulating Antioxidant Capacity and Stress Resistant Gene in Arabidopsis thaliana”;

Authors concentrated on the functions of drought and cold induced MbICE1 of M.baccata participating in Arabidopsis. The MbICE1 was transferred to Arabidopsis;

Moreover, under drought and cold conditions, the transgenic Arabidopsis had higher chlorophyll content and free proline content than WT plants, but the MDA content and electrolyte leakage were lower than those of WT plants. In addition, Predictably, Authors revealed that, drought and cold leaded to a large accumulation of ROS (H2O2 and O2- ) content in Arabidopsis, while overexpression of MbICE1 enhanced the antioxidant enzyme activity in Arabidopsis and finally improved the plant's resistance to stresses.

Obtained results seem to have potential and give valuable insight into plant abiotic stress response, but some aspects need to be explained as well as improved:

Introduction gives the reader sufficient background to analyze obtained results; Unfortunately, suddenly appears (line 41) ICE 1 – there is no smooth transition between above lines- Please, correct it;

But Please, explain what does it mean “CBF expression 1”;

Moreover, Please add the clearly underlined aim of the study – for example between lines 67/68;

Results – Figure 1 should be enlarge as much as it was possible, because the reader losses valuable information;

Moreover, Please add microphotographs in better quality in Figure 3A, B, D;

Please, underline, that the localization was done only in epidermis, not in plants other tissues;

Please, standarize fonts and sizes in the whole manuscript, because some paragraphs appear to be inserted and copied;

The ‘recover time’ should be added earlier, the reader finds it only at the end of methodology;

Please, enlarge Figure 4, because it is unable to even check which data is statistically significant;

The relative gene expression (4.4) in very good, valuable molecular studies should be analyze based on two, not only one, reference gene- like in 4.7 for example not only compared to actin but also to e.a. ubiquitin; Please, complete it;

Please explain: “With WT as the control, 6 (S1, S3, S4, S6, S8, and S9) lines can be identified by PCR gel electrophoresis  in 10 overexpression-MbICE1 lines (Figure 5A)”;

Furthermore, Please, fit Figure 6 to the whole page, because the obtained results are almost unreadable;

Sincerely

Author Response

I have an honor to review manuscript entitled “MbICE1 Confers Drought and Cold Tolerance through Up-regulating Antioxidant Capacity and Stress Resistant Gene in Arabidopsis thaliana”;

Authors concentrated on the functions of drought and cold induced MbICE1 of M.baccata participating in Arabidopsis. The MbICE1 was transferred to Arabidopsis;

Moreover, under drought and cold conditions, the transgenic Arabidopsis had higher chlorophyll content and free proline content than WT plants, but the MDA content and electrolyte leakage were lower than those of WT plants. In addition, Predictably, Authors revealed that, drought and cold leaded to a large accumulation of ROS (H2O2 and O2- ) content in Arabidopsis, while overexpression of MbICE1 enhanced the antioxidant enzyme activity in Arabidopsis and finally improved the plant's resistance to stresses.

Obtained results seem to have potential and give valuable insight into plant abiotic stress response, but some aspects need to be explained as well as improved:

We appreciate your valuable suggestions and advices on our manuscript.  I think they are very helpful and important, and revisions had been made in the revised manuscript accordingly.

Here I would like to response the comments and add some explanations as follows.

Supplementary materials and pictures have also been corrected.

Introduction gives the reader sufficient background to analyze obtained results; Unfortunately, suddenly appears (line 41) ICE 1 – there is no smooth transition between above lines- Please, correct it;

Response: Yes, we have accepted your suggestion and have changed this part.

But Please, explain what does it mean “CBF expression 1”;

Response: Gilmour et al. (1998) proposed for the first time that there is a inducer of CBF expression upstream of the cold induction signal in plant cells. And the inducer of CBF expression 1 is named ICE1.

Moreover, Please add the clearly underlined aim of the study – for example between lines 67/68;

Response: Yes, we have accepted your suggestion and have added our research aim.

Results – Figure 1 should be enlarge as much as it was possible, because the reader losses valuable information;

Response: Yes, we have accepted your suggestion and have enlarged Figure 1.

Moreover, Please add microphotographs in better quality in Figure 3A, B, D;

Response: Yes, we have accepted your suggestion and have replaced the photos with clearer images. 

Please, underline, that the localization was done only in epidermis, not in plants other tissues;

Response: Yes, we have accepted your suggestion. And in the method, the location of positioning is defined.

Please, standarize fonts and sizes in the whole manuscript, because some paragraphs appear to be inserted and copied;

Response: Yes, we have accepted your suggestion and we have corrected this part.

The ‘recover time’ should be added earlier, the reader finds it only at the end of methodology;

Response: Yes, we think it's a good suggestion. A recovery time of 7 days is mentioned in the results and methods.

Please, enlarge Figure 4, because it is unable to even check which data is statistically significant;

Response: Yes, we have accepted your suggestion and have enlarged Figure 4.

The relative gene expression (4.4) in very good, valuable molecular studies should be analyze based on two, not only one, reference gene- like in 4.7 for example not only compared to actin but also to e.a. ubiquitin; Please, complete it;

Response: Yes, we have accepted your suggestion and added a new internal reference gene and recalculated the relative expression level of MbICE1.

Please explain: “With WT as the control, 6 (S1, S3, S4, S6, S8, and S9) lines can be identified by PCR gel electrophoresis  in 10 overexpression-MbICE1 lines (Figure 5A)”;

Response: Yes, we have accepted your suggestion and we have explained in the paper.

Furthermore, Please, fit Figure 6 to the whole page, because the obtained results are almost unreadable;

Response: Yes, we have accepted your suggestion and have enlarged Figure 6.

Round 2

Reviewer 1 Report

The authors have addressed my concerns. They have made a substantial modification to the manuscript. I recommend it for publication.

Regards

Reviewer 2 Report

Dear authors,

Thanks for addressing my comments. I can bow recommend your manuscript for publication.

Best regards

Reviewer 3 Report

In my opinion Authors improved their manuscript in a significant way - point by point according my and  all Reviewer's suggestions, therefore I suggest to accept it in current improve form;

Sincerely